# Emergence of the structure-directing role of f-orbital overlap-driven covalency

Erli Lu[1], Saira Sajjad[1,2], Victoria E.J. Berryman[1], Ashley J. Wooles[1], Nikolas Kaltsoyannis [1] & Stephen T. Liddle [1]

FEUDAL (f's essentially unaffected, d's accommodate ligands) is a longstanding bonding model in actinide chemistry, in which metal-ligand binding uses 6d-orbitals, with the 5f remaining non-bonding. The inverse-trans-influence (ITI) is a case where the model may break down, and it has been suggested that ionic and covalent effects work synergistically in the ITI. Here, we report an experimentally grounded computational study that quantitatively explores the ITI, and in particular the structure-directing role of f-orbital covalency. Strong donor ligands generate a *cis*-ligand-directing electrostatic potential (ESP) at the metal centre. When f-orbital participation, via overlap-driven covalency, becomes dominant via short actinide-element distances, this ionic ESP effect is overcome, favouring a *trans*-ligand-directed geometry. This study contradicts the accepted ITI paradigm in that here ionic and covalent effects work against each other, and suggests a clearly non-FEUDAL, structure-directing role for the f-orbitals.

[1] School of Chemistry, The University of Manchester, Oxford Road, Manchester M13 9PL, UK. [2] Department of Chemistry, COMSATS Institute of Information Technology, Abbottabad, 22060 Abbottabad, Pakistan. These authors contributed equally: Erli Lu, Saira Sajjad, Victoria E.J. Berryman. Correspondence and requests for materials should be addressed to N.K. (email: nikolas.kaltsoyannis@manchester.ac.uk) or to S.T.L. (email: steve.liddle@manchester.ac.uk)

One of the most fascinating, enduring, and controversial topics in molecular actinide science is the continuously debated nature and extent of the chemical bonding of the early members of the series, and in particular uranium, and how this relates to structure and periodic trends within the context of the entire Periodic Table[1–19]. The role of s-, p-, and d-orbitals in chemical bonding and how this relates to the geometries of main group and transition metal complexes is now well understood. For lanthanides, the bonding is usually described as overwhelmingly ionic and non-directional with little orbital contribution; however, where covalency is invoked, for example where the *trans*-influence (TI) has been observed[20–30], it is usually d-orbitals that are involved[27,31,32]. By contrast, for the early actinides there is still debate over the extent and tensioning of f- vs d-orbital character[7,8,14] and, given that electrostatics are generally accepted as the dominant feature of the bonding, the structure directing role of the 5f-orbitals remains a moot point[33].

A longstanding conceptual bonding model in actinide chemistry is Bursten's FEUDAL (f's essentially unaffected, d's accommodate ligands). This model advances the notion that actinides bind primarily using their d-orbitals and the f-orbitals remain mainly non-bonding[34,35]. This view seems to hold for ions like uranium when bonded to expansive ligands that have low angular requirements, e.g. $C_{5–8}$-arenes[36–41], but in recent years this has been increasingly challenged when small ligands with more acute angular requirements, e.g. nitrides, are considered[42,43]. However, although this model considers the issue of orbital interactions it does not directly address whether f-orbitals are structure directing, but their characterisation as 'unaffected' implicitly suggests no structure-directing role. Moreover, there are hints in the literature that FEUDAL sometimes breaks down in scenarios where structure-directing effects involving f-orbitals are invoked. The longstanding, preeminent example of this is the inverse-*trans*-influence (ITI)[44–46], where strongly donating ligands are preferentially found to be *trans* to one another. The most prevalent example of this is uranyl; e.g. in $[UO_2Cl_2(OPPh_3)_2]$ the two oxos are mutually *trans* in direct contrast to $[MoO_2Cl_2(OPPh_3)_2]$ where they are *cis*[47–50]. In recent years a variety of non-uranyl complexes that seem to exhibit the ITI have been reported, usually with uranium in oxidation states V and VI and with nitride or oxo ligands[43,51–56]. Two hypotheses have been developed to account for the ITI[45,46,57–59]. From an orbital perspective, it is proposed that the 6p-orbitals of early actinides are semi-core, and therefore semi-valence, and can donate electron density into vacant 5f-orbitals; thus, an electron hole forms that is compensated by increased electron donation from a trans ligand. Alternatively, a polarisation argument can be employed; when the parity of overlapping orbitals is *u-g*, as is the case with p-d orbitals, a dipolar arrangement at the metal disfavours *trans* and stabilises *cis* positions, whereas when the parity is the same, e.g. *u-u* for p-f orbitals, then the charge distribution is quadrupolar with *trans* favoured and *cis* disfavoured. It should be noted that these working theories are based on logical, but suppositional, arguments, and although the majority of studies have focussed on establishing the role of the 6p- and/or 5f-orbitals, their precise roles remain somewhat nebulous. Furthermore, although the traditional view of the ITI is that ionic and covalent effects compete with one another, it has been suggested that this is misleading[33], and that ionic and covalent effects are actually working together synergistically.

At this point, a clarification of the term covalency is merited[60–62]. Covalency, that is the mixing coefficient, is proportional to the spatial overlap of the orbitals divided by the difference in their energies, and these two parameters are independent of one another[7,8]. So, covalency can increase by increased spatial overlap or by reduction in the difference of parent atomic orbital energies.

The latter, which is called near-energy driven covalency, is a perfectly valid definition of covalency when framed in context, but chemical bonding carries the connotation of orbital overlap resulting in electron density building up in the inter-nuclear region. The former is called overlap-driven covalency, and it is on this aspect that discussions in this paper will focus.

We recently reported, Fig. 1, an extension of the ITI to tetravalent cerium, uranium, and thorium *trans bis*(carbene) complexes (**1Ce**, **1U**, **1Th**)[62] and latterly found that in carbene-imido derivatives of uranium, *cis* geometries were overwhelmingly favoured (**2UNHRK** and **2UBIPY**)[63,64]. It is important to note that in the former the *trans* disposition is enforced by ligand steric constraints yet the *trans* carbenes are strongly bound to the metals with short M=C bond distances, but in the latter even when steric constraints are removed the *cis* geometry dominates. Since uranium often uses more 5f- than 6d-orbital character in its bonding, but the reverse is usually found for thorium, we prepared analogous thorium carbene-imido complexes. Again, a *cis* geometry is preferred, which spurred us to survey the inherent *cis* or *trans* preferences for carbene-carbene, carbene-imido, and carbene-oxo ligand combinations for cerium, uranium, and thorium. Though many systems indeed prefer *cis* geometries, we notably find that the cerium and uranium oxo systems go against this trend and in fact prefer *trans* geometries. This study puts the original proposition of the ITI on a quantified, firm footing and reveals that the electrostatic potential (ESP) surface around the metal centre directs the *cis* geometries. This first concerted application of ESP arguments to this issue reveals that d-orbital participation is in fact not the driving force for *cis* geometries, but opportunistically results from ESPs. When f-orbital participation, and associated overlap-driven covalency, becomes dominant, the ionic effect is overcome and a *trans* geometry is favoured. This study therefore demonstrates that ionic and covalent effects work against each other in cases where the structure-directing role of f-orbitals is confirmed, the latter aspect challenging the generality of the FEUDAL model.

## Results

**Synthesis, characterisation, and solid state structures**. With **1M** (M = Ce, U, Th) and **2UNHRK** and **2UBIPY** reported[62–64], we sought to prepare the analogous $R_2C = Th^{IV} = NCPh_3$ (**2Th**, R = $Ph_2PNSiMe_3$) complexes, in order to now map out the TI/ITI structural influences across the $C = M^{IV} = E$ (E = $CR_2$, $NCPh_3$) series, Fig. 2. It is noteworthy that in comparison with the burgeoning nature of uranium-ligand multiple bond species[3,6,12–14], thorium-ligand multiple bond species are less developed. Indeed, for thorium 2-metalla-allenes, despite their significant importance as relatives of $ThO_2$, two homoleptic thorium-*bis*(carbenes) are the only such species in the literature[62,65] and there are no heteroleptic thorium-2-metalla-allenes. The previously reported thorium-carbene-*bis*(alkyl) $[Th^{IV}(BIPM^{TMS})(CH_2SiMe_3)_2]$ (**3**)[62], which has a pre-installed Th=C double bond interaction, was found to be a suitable precursor to preparing $C = Th^{IV} = N$ linkages, Fig. 2 (see Supplementary Information). Complex **3** is straightforwardly converted, via the diamide (**4**) or alkyl-amide (**5**) into **2ThBIPY** or **2ThNHRK**, respectively, which bear the desired $C = Th^{IV} = N$ unit, via two-step syntheses that involve deprotonations facilitated by external (for **2ThNHRK**) or internal (for **2ThBIPY**) Brønsted bases, and these complexes are isolated as red crystalline solids in satisfactory yields. The formulations of **2ThNHRK** and **2ThBIPY** are supported by NMR, IR, and optical spectroscopies, and elemental analyses (see Supplementary Figures 1 to 14).

Beyond the spectroscopic data, the structures of **2ThNHRK** and **2ThBIPY** are unambiguously confirmed by X-ray single

**Fig. 1** Carbene complexes. Previously reported work[63,64] and *trans* and *cis* models studied in this work

**Fig. 2** Synthesis of complexes **2ThBIPY** and **2ThNHRK**. The known dialkyl complex **3** can be reacted two equivalents of trityl-amine by protonolysis to give the diamide complex **4**. Complex **4** when treated with two equivalent of benzyl potassium (to maximise the yield, a stoichiometric amount of benzyl potassium gives lower yields) converts to the amide-imide complex **2ThNHRK**. Complex **3** can alternatively be reacted with a sub-stoichiometric quantity of trityl-amine (to suppress ligand-redistribution reactions) to give, via protonolysis, the mixed alkyl-amide complex **5**. Complex **5** undergoes α-hydrogen abstraction on addition of BIPY to give **2ThBIPY**

crystal diffraction, Fig. 3. The salient structural feature of these complexes is the *cis*-C=Th$^{IV}$=N units [C=Th=N for **2ThNHRK** and **2ThBIPY** = 107.08(19) and 110.90(9)°, respectively], which is similar to **2U** analogues, suggesting the presence of a TI. Structurally speaking, though a potassium ion is intimately coordinated in the structure of **2ThNHRK** the Th=N$_{imide}$ and Th=C$_{carbene}$ bonds in this complex are little disturbed from what might be anticipated for formal thorium-nitrogen and -carbon double bond interactions [Th=N$_{imide}$ for **2ThNHRK** and **2ThBIPY** = 2.109(5) and 2.067(2) Å; Th=C$_{carbene}$ for **2ThNHRK** and **2ThBIPY** = 2.564(6) and 2.558(3) Å, respectively]. The Th–N$_{BIPY}$, C–N and C–C bond lengths in the bipyridine fragment of **2ThBIPY** are consistent only with a neutrally coordinated BIPY ligand[66].

**Computational geometry optimisations**. Building on our previous study of **2UBIPY**[64], we investigated a family of nine metalla-allene model systems [C]=M=E ([C]=C(PH$_2$NSiH$_3$)$_2$; M = Ce$^{IV}$, Th$^{IV}$, U$^{IV}$; E = C(CH$_3$)$_2$, NCH$_3$, O), with a particular focus on the C-M-E angle, using the Gaussian-09 code[67] with two density functional approximations (DFAs). We used the generalised gradient approximation (GGA), PBE[68,69], and related hybrid, PBE0[70]; these DFAs are ideal as PBE has recently been shown to give accurate geometries in an extensive benchmarking study of organouranium systems[71], and the GGA BP86 performs better than B3LYP and certain Minnesota functionals for some uranium bis carbene complexes[72], and PBE0 is known to give improved energetics and has been

previously applied by us to the study of a uranium(IV)-carbene-imido complexes[63,64]. Model complexes were sterically truncated and void of potassium ions and co-ligands to isolate electronic effects from steric constraints, and the final equilibrium geometries are obtained irrespective of whether the starting geometry is *cis* or *trans* with respect to the [C]=M=E angle. The results are collected in Supplementary Tables 1 to 5, from which it can be seen that there is little difference between the two DFAs. All the systems with E=C(CH$_3$)$_2$ and NCH$_3$ adopt a *cis* geometry. However, for the oxo complexes only [C]=Th=O has a *cis* conformation (C-Th-O angle = 116.8/116.6°), whereas [C]=Ce=O and [C]=U=O prefer *trans* geometries, with C-M-O angles of 165.1/162.2° and 176.9/176.4°, respectively, at the PBE0/PBE level.

**Computational total energy surface scans**. To further probe the energetic preference for *cis* or *trans* conformations, total self-consistent field (SCF) energy surfaces were explored as a function of the C-M-E angle, as defined by the *trans* and *cis* models on the right-hand side of Chart 1. All geometric parameters were relaxed except this angle, which was perturbed in 5° increments from the optimised geometry. The resulting plots for the E=C(CH$_3$)$_2$, NCH$_3$ and O systems are shown in Fig. 4a–c, respectively. The data for [C]=M=C(CH$_3$)$_2$ and [C]=M=NCH$_3$ are similar to one another; in both cases the Th molecule has the largest preference for a *cis* geometry, followed by U and then Ce, which are similar. We have attempted to quantify these preferences by locating transition states (TSs), starting from the highest points of the SCF

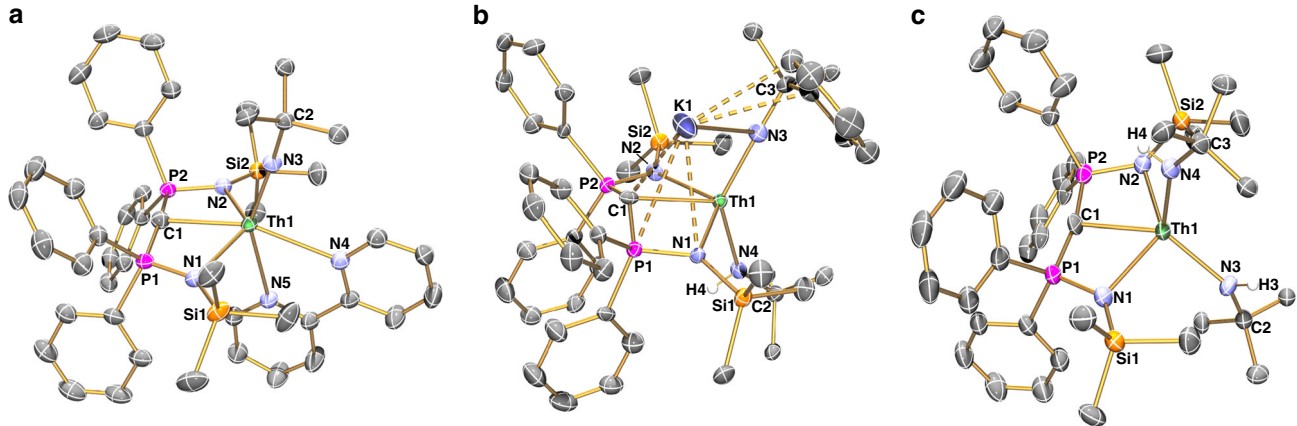

**Fig. 3** Molecular structures of the molecules reported in this study. **a 2ThBIPY**, **b 2ThNHRK**, and **c 4**. Structures were determined at 120 K and are shown with displacement ellipsoids set to 40%. Hydrogen atoms, minor disorder components, lattice solvent, and non-*ipso* trityl-phenyl ring carbon atoms, unless involved in a novel binding interaction, are omitted for clarity

energy scans; the available data are presented in Supplementary Table 2. In [C]=Th=NCH₃, the maximum of the energy surface scan is at 164.7°, and this point is 27.1 kJ mol⁻¹ above the optimised structure (Fig. 4b). A coupled cluster (CCSD(T)) single-point calculation at the PBE0 geometry of the maximum energy point lies 27.0 kJmol⁻¹ above that of the fully optimised geometry, providing excellent post-Hartree–Fock validation of the PBE0 approach. The very small T1 diagnostic (0.017) at both the fully optimised and maximum energy point structures indicates that the electronic structure is well represented by a single configuration, providing further justification of a DFT-based analysis.

In contrast to the C- and N-based systems, the total energy scans for the [C]=M=O model complexes (Fig. 4c), reveal the Ce and U molecules to be most stable at large C-M-O angles. [C]=U=O is particularly interesting, with a shallow local minimum at a C-U-O angle of ~122°, which interestingly is close to the lowest energy optimised geometry of [C]=Th=O. The transition state for the *cis-trans* isomerisation for this complex (at 142.2°) is only 3.3 kJ mol⁻¹ above the fully optimised structure. For [C]=Ce=O, there is no local minimum at a *cis* geometry; however, reducing the angle to 120° incurs an energy change of less than 5.0 kJ mol⁻¹. Similarly, the *cis/trans* isomerisation barrier for [C]=Th=O is reduced by approximately half relative to [C]=Th=C(CH₃)₂ and [C]=Th=NCH₃, to only 14.7 kJ mol⁻¹. Thus, all three model oxo systems have a significant reduction in the energy required to manipulate the C-M-E angle in comparison with the C- and N-based systems.

**Electrostatic potential analysis.** The above data show that for seven of the nine model complexes examined, the *cis* geometry is clearly intrinsically favoured, whereas for two of them, namely [C]=M=O (M=U, Ce) a *trans* geometry is preferred. In order to probe and understand why this is the case, we conducted electrostatic potential (ESP) calculations coupled to the commonly employed natural localised molecular orbital (NLMO) approach[73]. Electrostatic potentials allow for the visualisation of the charge distribution of a molecule. ESPs are used extensively to understand complex systems, such as enzymes, but have not hitherto been employed in actinide chemistry outside our previous study[64].

Previously, we advanced an explanation for the bent structure of [C]=U=NCH₃ based on the *cis*-directing nature of the ESP of the [[C]=U]²⁺ fragment[64]. The ESP surface around the metal is asymmetric, and favours a negatively charged E ligand at the *cis* position. To probe the generality of this effect, we have now

conducted analogous calculations on [[C]=Th]²⁺, and the results are shown in Fig. 5. This shows the evolution of the ESP surface (positive everywhere for a dicationic system) as the [C] ligand is brought up to the Th in 1 Å steps, starting from a point at which the Th-C₍C₎ distance is 4 Å longer than in the optimised geometry of [C]=Th=NCH₃. As with [[C]=U]²⁺, at long r(Th-C₍C₎) the ESP around the Th is essentially isotropic, but as r(Th-C₍C₎) shortens pronounced anisotropic character develops, with the region of positive ESP extending towards the position that the E ligands occupy in [C]=Th=E. Thus, as for [[C]=U]²⁺, the interaction of the BIPM model with the Th generates an ESP which is *cis*-directing towards an incoming negatively charged ligand.

**Natural localised molecular orbital analysis.** In order to understand the origin of the asymmetric ESP, we analysed the NLMOs of [[C]=Th]²⁺ at the equilibrium r(Th-C₍C₎) distance. The NLMO shown in Fig. 6 is highly directional, and would clearly disfavour an incoming negatively charged ligand in the *trans* position, i.e. there would be substantial repulsion between a ligand approaching from the right hand side of Fig. 6 and the electron in the NLMO shown. This likely accounts for the *cis*-directing ESP. The asymmetric ESP of [[C]=M]²⁺ nicely explains the small [C]-M-E angles in the majority of the [C]=M=E models studied here. However, the near linearity of [C] = Ce = O and [C] = U = O is not consistent with that observation, which means that there must be another effect at work for those two systems. We therefore conducted an NLMO analysis of all nine [C] = M = E systems, particularly focusing on the M=E bonding orbitals. Comparison of these at the optimised geometries with those of the structures at the end of the energy scans reveals that the σ-orbitals exhibit the greatest energy changes; at least twice those found for the π-orbitals and more often substantially greater (~×20). This data can be found in Supplementary Table 3. Additionally, the σ-orbitals exhibit the greater change in contribution from the metal; on average, these changes are an order of magnitude larger in the σ- than the π-orbitals. Thus, we focus our analysis on the metal-ligand σ-bonding NLMOs, composition data for which are collected in Table 1, and a representative example of which is shown in Supplementary Figure 15.

For all three E ligands, the total metal contribution to the M–E σ-bonding NLMO is ordered Ce ≈ U > Th. This metal contribution is predominantly d-character, except for the [C]=Ce=O and [C]=U=O systems, for which the dominant metal contribution comes from the f-orbitals. Figure 7 plots the total f-contribution

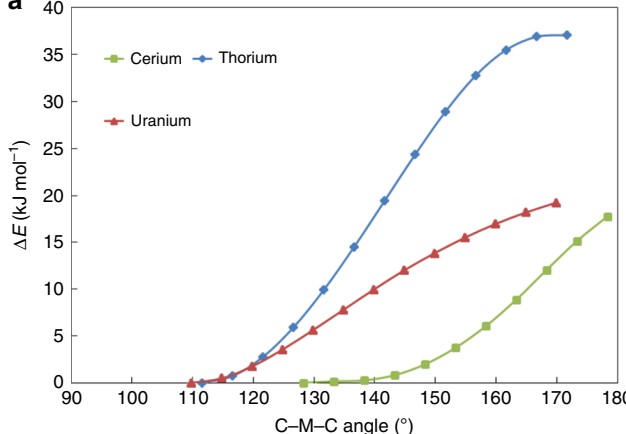

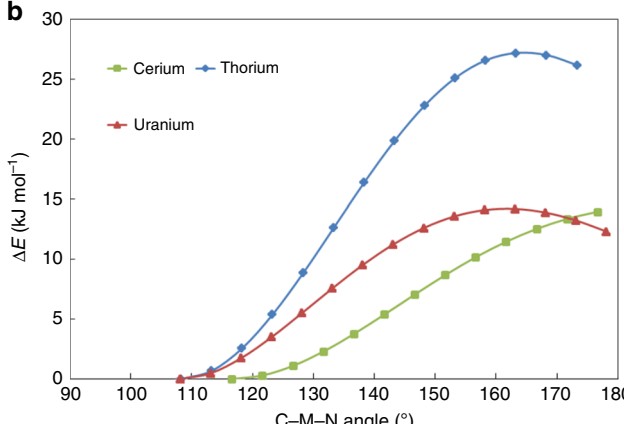

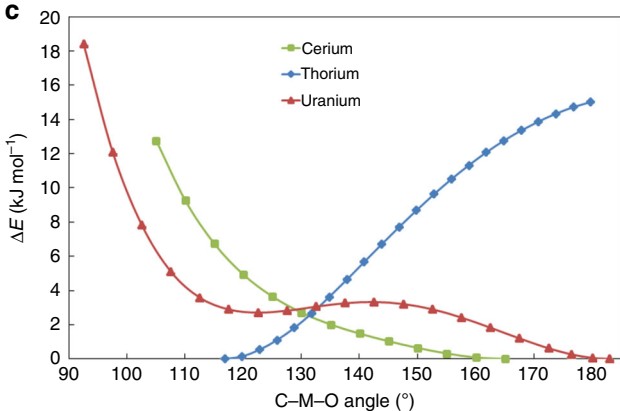

**Fig. 4** PBE0 SCF energy surface scans of [C]=M=E angles, M=Ce, Th, U. **a** E=C(CH$_3$)$_2$, **b** E=NCH$_3$, **c** E=O. Energies (kJ mol$^{-1}$) are presented relative to those of the fully optimised structures

reduce the effect. We therefore moved the model BIPM ligand away from the metal centre by extending the distance between the metal and the central C atom of [C] by 1 Å, and probed the effect on the energy barrier to alteration of the C-M-E angle. As expected, these barriers are either reduced significantly or removed altogether; the effect on the [C]=Th=C(CH$_3$)$_2$ system is shown in Supplementary Figure 16. For this molecule, the barrier is reduced by more than half. Extending this argument, elongating the M-O distance in either [C]=Ce=O or [C]=U=O should reduce the M-O interaction and favour a smaller C-M-O angle. This was probed by lengthening the Ce-O distance from its optimised value of 1.77 to 3.0 Å, and indeed the preferred C-Ce-O angle decreases from 165 to 135°.

For all seven C-M-E bending TSs located (Supplementary Table 4), there is very little change in $r$(M-E), but in all cases bar [C]=U=O there is significant elongation (>0.08 Å) of $r$(M-C$_{[C]}$) at the TS. This lengthening, and presumably weakening, of the M-C$_{[C]}$ interaction destabilises the TS vs the true minimum geometry. By contrast, the changes in both $r$(M–E) and $r$(M–C$_{[C]}$) at the bending TS located for [C]=U=O are very modest, in agreement with this TS being of much lower relative energy than the other six. Indeed, the energy surfaces presented in Fig. 4a–c suggest that the TS for [C]=U=O could be considered separately from the rest.

The classic *trans* influence in transition metal element chemistry arises from the competition for metal d-orbitals between two mutually *trans* ligands, resulting in the elongation of the bond *trans* to the stronger donor ligand. If this were a key factor in our systems, we would expect the lengthening of $r$(M–C$_{[C]}$) to be related to the d-orbital content of the M-E NLMO at the TS. However, we find essentially no such correlation between these variables; $R^2 = 0.20$ for the correlation of the total metal d-orbital contribution to the TS' M-E σ NLMO with the $r$(M-C$_{[C]}$) elongation (in the six TSs bar that in [C]=U=O). An alternative explanation for the $r$(M-C$_{[C]}$) elongation is an extension of the electrostatic argument presented above; rotating the E ligands away from the optimised C-M-E angles and towards linearity moves them from the orientation favoured by the asymmetric ESP surface around the metal, and the system adjusts by attempting to reduce the asymmetric ESP by elongating $r$(M-C$_{[C]}$).

GGA-type DFAs typically favour greater electron delocalisation, leading to more radially diffuse orbitals. It is interesting to note that for almost all of the complexes studied here the GGA-type functional, PBE, produces σ-bonding NLMOs with greater f-orbital character (Table 1). The only exception is [C]=U=NCH$_3$, which shows a significant increase in s orbital contribution. However, it is important to note that the differences between PBE and PBE0 are slight, and a consistent trend emerges whereby the f-orbital contribution is, like-for-like, always greater than the d-orbital contribution for Ce and U compared to Th. This is the case irrespective of the identity of E, but is certainly most pronounced for the oxo complexes, giving confidence that the high f-orbital contributions to the Ce=O and U=O bonds are real and not a computational artefact.

**Topological bonding analysis.** To further investigate the bonding between the metal centre and E ligand, analysis of the topology of the electron density was carried out with the Quantum Theory of Atoms in Molecules (QTAIM)[74,75]. We focus on three parameters; the electron density at the bond critical point (BCP) along the bond path between the M and C/N/O atomic centres ($\rho_{BCP}$), the total energy density at the BCP ($H_{BCP}$) and the delocalisation index between the M and C/N/O atomic basins $\delta$(M,E). The magnitudes of these parameters, in an absolute sense,

to the M-E σ-bonding NLMO against the [C]=M=E angle for all nine targets; there is a striking correlation, with $R^2 = 0.88$ at the PBE0 level, indicating that the larger the total f-orbital contribution to the σ-bonding NLMO the larger the [C]=M=E angle, i.e. the [C]=M=E angle is a function of f-orbital based overlap-driven covalency. In contrast, there is very little correlation of the [C]=M=E angle with metal d-character to the NLMO σ-bonding ($R^2 = 0.36$ at the PBE0 level).

If indeed the interaction of the (model) BIPM ligand with the metal centre directs the E ligand into the *cis* position due to the asymmetric ESP, it would be expected that reducing that interaction by moving the [C] away from the metal would

**Fig. 5** Electrostatic potential (ESP) surface of the $[C]=Th^{2+}$ fragment. As a function of distance this shows the approach of the model BIPM [C] ligand to the Th $[r(Th-C_{[C]})]$ at **a** 6.491 Å, **b** 5.491 Å, **c** 4.491 Å, **d** 3.491 Å, **e** 2.491 Å, the value in the optimised structure of $[C]=Th=NCH_3$. The isovalue is 0.5

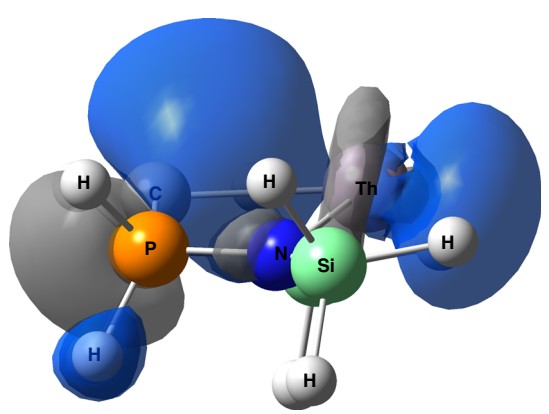

**Fig. 6** NLMO of $[C]=Th^{2+}$. This NLMO has significant amplitude in the *trans* direction, resulting in a *cis* directing effect. The orbital is 16.86% Th character, and that component is composed of 19.77% 7s, 63.88% 6d, and 16.05% 5f

provide a measure of the overall extent of covalency in the bonding interactions. The data for the nine model complexes are summarised in Table 2, and reveal some clear trends. For a given metal, all three metrics increase (in an absolute sense) in the order $C(CH_3)_2 < NCH_3 < O$ while, for a given E ligand, the QTAIM metrics increase in the order Th < U < Ce. The data indicate that $[C]=Ce=O$ and $[C]=U=O$ have the most covalent M-E interactions, in agreement with the suggestion that the *cis*-directing ESP is overcome only in the most covalent of our systems.

The QTAIM metrics give us a measure of overall covalency, whereas NLMO analysis allows us to assess specific orbitals. Together they provide complementary methods to assess covalency and, ideally, we expect correlations between the data from the two techniques. This is assessed in Table 3, where the regression analyses for the correlation of a number of key variables are presented. In all bar three cases, the $R^2$ values are well over 0.9, indicating strong correlations between the NLMO and QTAIM metrics. This is particularly so for $\rho_{BCP}$. That both orbital and electron density-based assessments of covalency correlate so well gives confidence in our conclusions regarding the extent of covalency in these An–E interactions.

## Discussion

Noting the orbital and parity arguments for the ITI presented above, we recognise that either of these logical, but largely speculative, arguments can be combined and subsumed into the ESP argument based on the calculation on the electronic structure of $[[C]=M]^{2+}$. It is clear from the computed data that the thorium complexes have much the strongest preference for a *cis* geometry. At first, if considering orbital arguments, this is counterintuitive because the bonding of thorium is more ionic than uranium and so orbital factors, and thus the *cis* effect, should be diminished.

However, when an ESP argument is considered, the stronger preference of thorium to adopt a *cis* geometry falls entirely into line with what would be predicted based on where the charge build-up occurs, i.e. *trans*, thus leaving a *cis* hole to accommodate a *cis* ligand. This might be linked to thorium d-orbital character, however although the computed data in Table 1 superficially supports this, more detailed assessment shows that this is not the case. This leads us to an important conclusion, which is that in the absence of other drivers it is the ESP that dominates the resulting geometry; this is not contingent on the d-orbital character in the M-E bond, but this does not mean that d-orbitals may not be used as a consequence. So, d-orbital character may result from the *cis* geometry but the *cis* geometry does not itself result from d-orbital character. This ESP argument thus extends and refines Denning's original proposition[45,46] into a more quantified, and firmer, basis.

Having established that the preferred geometry of the complexes in this study is *cis*, we now address why $[C]=U=O$ and $[C]=Ce=O$ prefer *trans* geometries. We propose that the optimised $[C]=M=E$ angles arise from the interplay of electrostatic (ionic) and orbital (covalent) effects; the former favour the *cis* orientation while the latter favour linearity. It is likely that the orbital effects dominate in $[C]=Ce=O$ and $[C]=U=O$ because of the small size of $O^{2-}$; these two systems have the shortest M–E distances (Supplementary Table 4) and hence only in $[C]=Ce=O$ and $[C]=U=O$ is the M–E distance short enough to allow sufficient f-orbital/ligand overlap for the covalent driver to linearity to overcome the *cis*-directing ionic effect. Certainly, f-character dominates the M-E σ-bonding NLMOs of $[C]=U=O$ and $[C]=Ce=O$ in contrast to the other seven model systems where d-character dominates, Table 1. Looking more widely, it is certainly the case that where the ITI clearly occurs or is proposed to occur this almost always involves small, highly charged ligands such as $N^{3-}$ and $O^{2-}$ with short M–E distances[43,51–57]. In other words, given the radially contracted nature of 5f orbitals compared to 6d, only at short M–E distances can the 5f-orbitals come into the bonding picture and exert their influence to favour a *trans* $[C]=M=O$ geometry. As we noted earlier, the oxo systems require significantly smaller energies to manipulate the C–M–E angle in comparison with the C- and N-based systems, and we suggest that this is because the ionic and covalent effects are most finely balanced in these molecules. For $E=C(CH_3)_2$ and $NCH_3$, Th has much the strongest preference for the *cis* geometry, and $[C]=Th=O$ is the only oxo to favour significantly bent C-Th-O. For a given E ligand, the Th compound has the lowest f contribution to the M-E σ-bonding NLMO, and the QTAIM metrics are the smallest. Hence the Th–E interaction is clearly the most ionic and its geometry is dominated by electrostatics. By contrast, for $[C]=M=O$ (M=Ce, U) f-orbital covalency plays a structure-dictating role, something normally (i.e. in the FEUDAL model) limited to d-orbitals. Indeed, even for lanthanides TI effects have been convincingly attributed to the role of d-, not f-, orbitals in bonding to ligands[27,31]. We therefore suggest that, within the interplay of early actinide ionic vs covalent effects, the structure-directing capacity of overlap-driven covalency is not solely the domain of the d-orbitals.

**Table 1 Composition (%) of the M–E σ bonding NLMOs of [C]=M=E ([C]=C(PH₂NSiH₃)₂; M=Ce, Th, U; E=C(CH₃)₂, NCH₃, O) at their optimised geometries[a]**

| E | M | Functional | Contribution of M to the σ-bonding NLMO | | | | | Total d | Total f |
|---|---|---|---|---|---|---|---|---|---|
| | | | M | s | p | d | f | | |
| C(CH₃)₂ | Ce | PBE0 | 31.19 | 1.48 | 0.09 | 63.74 | 34.66 | 19.88 | 10.81 |
| | | PBE | 30.41 | 2.84 | 0.09 | 60.05 | 37.00 | 18.26 | 11.25 |
| | Th | PBE0 | 23.09 | 7.19 | 0.44 | 78.05 | 14.31 | 18.02 | 3.30 |
| | | PBE | 24.17 | 8.29 | 0.47 | 75.73 | 15.50 | 18.30 | 3.75 |
| | U | PBE0 | 28.89 | 8.32 | 0.24 | 61.19 | 30.25 | 17.68 | 8.74 |
| | | PBE | 30.56 | 10.59 | 0.21 | 56.83 | 32.36 | 17.37 | 9.89 |
| NCH₃ | Ce | PBE0 | 19.58 | 3.75 | 0.14 | 69.97 | 26.13 | 13.70 | 5.12 |
| | | PBE | 18.61 | 4.65 | 0.17 | 67.37 | 27.80 | 12.54 | 5.17 |
| | Th | PBE0 | 15.36 | 6.57 | 0.95 | 75.09 | 17.35 | 11.53 | 2.66 |
| | | PBE | 15.86 | 7.13 | 1.07 | 73.30 | 18.48 | 11.63 | 2.93 |
| | U | PBE0 | 19.18 | 7.91 | 0.28 | 60.01 | 31.77 | 11.51 | 6.09 |
| | | PBE | 19.67 | 11.25 | 0.32 | 59.94 | 28.48 | 11.79 | 5.60 |
| O | Ce | PBE0 | 26.60 | 3.04 | 0.38 | 32.90 | 63.53 | 8.75 | 16.90 |
| | | PBE | 26.47 | 3.02 | 0.45 | 31.33 | 65.14 | 8.29 | 17.24 |
| | Th | PBE0 | 16.11 | 1.48 | 4.92 | 59.70 | 33.77 | 9.62 | 5.44 |
| | | PBE | 16.80 | 1.73 | 5.32 | 56.38 | 36.49 | 9.47 | 6.13 |
| | U | PBE0 | 24.20 | 1.73 | 0.59 | 22.84 | 74.79 | 5.53 | 18.10 |
| | | PBE | 23.94 | 1.42 | 0.58 | 22.44 | 75.52 | 5.37 | 18.08 |

[a]The total d and total f values are the products of the metal contribution and its d and f breakdowns

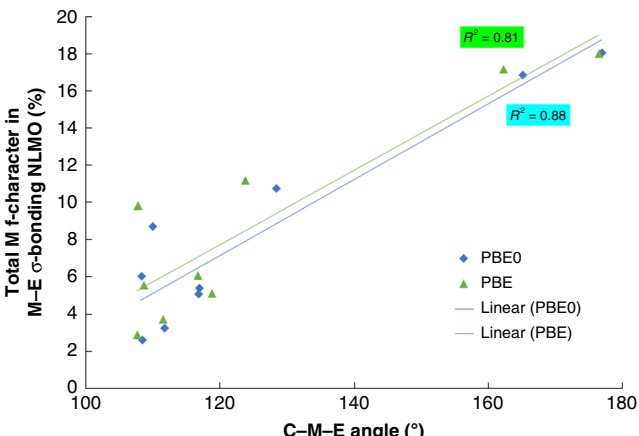

**Fig. 7** The C-M-E angle (°) vs the total metal f character in the M-E σ bonding NLMO (%) of [C]=M=E. C=C(PH₂NSiH₃)₂; M=Ce, Th, U; E=C (CH₃)₂, NCH₃, O). $R^2$ with PBE0 (PBE) for Ce, Th, and U are 0.93 (0.83), 0.97 (0.96), and 0.96 (0.88), respectively

**Table 2 QTAIM properties (PBE0) for the M–E interaction in [C]=M=E ([C]=C(PH₂NSiH₃)₂; M=Ce, Th, U; E=C(CH₃)₂, NCH₃, O)[a]**

| E | M | $\rho_{BCP}$ | $H_{BCP}$ | $\delta(M,E)$ |
|---|---|---|---|---|
| C(CH₃)₂ | Ce | 0.166 | −0.093 | 1.56 |
| | Th | 0.146 | −0.078 | 1.39 |
| | U | 0.156 | −0.082 | 1.48 |
| NCH₃ | Ce | 0.195 | −0.123 | 1.89 |
| | Th | 0.178 | −0.110 | 1.63 |
| | U | 0.197 | −0.125 | 1.80 |
| O | Ce | 0.284 | −0.247 | 1.91 |
| | Th | 0.242 | −0.202 | 1.68 |
| | U | 0.278 | −0.242 | 1.87 |

[a]$\rho_{BCP}$ is the bond critical point between the M and C/N/O centres, $H_{BCP}$ is the total energy density at that bond critical point, and $\delta(M,E)$ is the delocalisation index between the M and C/N/O atomic basins

To summarise, we have prepared thorium–carbene–imido complexes, which together with uranium analogues has enabled us to conduct an experimentally grounded computational study into TI and ITI effects in carbene-carbene, carbene-imido, and carbene-oxo ligand combinations at uranium, thorium, and cerium. By conducting calculations on models freed from steric and counter-ion constraints, we have been able to place the ITI on a quantified, firmer footing. We find that a strong donor ligand such as the carbene generates an ESP that is inherently *cis*-directing in terms of subsequent ligand coordination. This reveals that d-orbital participation in the M–E bonds may opportunistically result from this ESP but does not drive it. When f-orbital participation, with associated overlap-driven covalency, becomes dominant via short M–E distances then this ionic effect is overcome and a *trans* geometry is favoured. This study therefore contradicts the previous assessment of the ITI as resulting from synergistic interplay of ionic and covalent effects in that here the data suggest that they work against each other in cases where the structure-directing role of f-orbitals is confirmed. This work therefore also suggests an instance where FEUDAL breaks down. The structure-directing capacity of overlap-driven covalency would therefore seem to be not solely the domain of the d-orbitals; here the suggestion of the structure-directing role of f-orbital overlap-driven covalency emerges.

## Methods

**Preparation of [Th{C(PPh₂NSiMe₃)₂}(NHCPh₃)₂] (4).** At −78 °C, a solution of Ph₃CNH₂ (0.830 g, 3.2 mmol) in toluene (10 ml) was added to a stirring solution of **3** (1.926 g, 2 mmol) in toluene (10 ml). The mixture was allowed to stir at −78 °C for 30 min and at ambient temperature for 3 h. After which, the mixture was filtered, and all volatiles in the filtrate were evaporated under vacuum to afford a viscous yellow oil. The oil was washed with pentane (5 ml × 4) and dried under vacuum to afford **4** as a yellow solid. Yield: 1.400 g, 54%. Single crystals suitable for X-ray diffraction were obtained from toluene solution at 0 °C. Anal. Calcd for C₆₉H₇₀N₄P₂Si₂Th·0.5(C₇H₈): C, 64.43; H, 5.52; N, 4.15. Found: C, 63.64; H, 5.60; N, 3.64. ¹H NMR (C₆D₆, 298 K): δ (ppm) 0.01 (s, 18 H, -SiMe₃), 4.03 (s, 2 H, -NHCPh₃), 6.95–6.99 (m, 9 H, ArH), 7.02–7.08 (m, 13 H, ArH), 7.18–7.20 (m, 9 H,

**Table 3 Correlation ($R^2$) between QTAIM and NLMO properties (PBE0) of the M–E interaction in [C]=M=E ([C]= C(PH$_2$NSiH$_3$)$_2$; M=Ce, Th, U; E=C(CH$_3$)$_2$, NCH$_3$, O)[a]**

| Variables | | $R^2$ | | |
|---|---|---|---|---|
| | | C(CH$_3$)$_2$ | NCH$_3$ | O |
| Total M character in σ bonding M-E NLMO (%) | vs $\rho_{BCP}$ | 0.941 | 0.967 | 0.950 |
| | vs $H_{BCP}$ | 0.773 | 0.956 | 0.943 |
| | vs $\delta(M,E)$ | 0.956 | 0.933 | 0.918 |
| Total M f orbital character in σ bonding M-E NLMO (%) | vs $\rho_{BCP}$ | 0.937 | 0.967 | 0.953 |
| | vs $H_{BCP}$ | 0.766 | 0.976 | 0.965 |
| | vs $\delta(M,E)$ | 0.953 | 0.657 | 0.939 |

[a]$\rho_{BCP}$ is the bond critical point between the M and C/N/O centres, $H_{BCP}$ is the total energy density at that bond critical point, and $\delta(M,E)$ is the delocalisation index between the M and C/ N/O atomic basins

ArH), 7.54–7.59 (m, 19 H, ArH). $^{31}$P NMR (C$_6$D$_6$, 298 K): δ (ppm) 4.70 (s). $^{13}$C {$^1$H} NMR (C$_6$D$_6$, 298 K): δ (ppm) 3.91 (s, -SiMe$_3$), 77.81 (s, -NHCPh$_3$), 126.90, 128.78, 129.00, 129.46, 130.10 (ArC), 131.69 (t, $^2J_{PC}$ = 5.0 Hz, C$_{meta}$ of P–Ph), 139.72 (t, $^1J_{PC}$ = 49.1 Hz, C$_{ipso}$ of P–Ph), 152.20 (ArC). ATR-IR ν cm$^{-1}$: 3052 (w), 3019 (w), 2947 (w), 2892 (w), 1594 (w), 1488 (m), 1435 (s), 1346 (s), 1282(s), 1246 (m), 1177 (s), 1149 (m), 1105 (m), 1080 (s), 1043 (s), 1024 (s), 831 (m), 764 (m), 695 (s), 637 (s), 603 (m), 545 (m), 526 (s), 509 (s), 471 (s), 457 (s), 410 (s).

**Preparation of [Th{C(PPh$_2$NSiMe$_3$)$_2$}(=NCPh$_3$)(–NHCPh$_3$)(K)] (2ThNHRK).** At ambient temperature, 15 ml of benzene was added to a stirring solid mixture of **4** (652.3 mg, 0.5 mmol) and KBn (136.7 mg, 1.05 mmol) to afford a brick red suspension. The mixture was stirred at ambient temperature for 3 h and filtered. All volatiles were removed from the red solution, the red residue was washed with pentane (5 ml × 5) and dried in vacuo to afford **2ThNHRK** as a red solid (385.2 mg, 57%). Single crystals suitable for X-ray diffraction were obtained from benzene solution under ambient temperature. Once obtained as crystalline material, **2ThNHRK** is not soluble in aromatic and aliphatic solvents, and decomposes in coordinative and polar solvents. So, the $^1$H and $^{31}$P NMR spectra were recorded from the NMR scale reaction. However, satisfactory $^{13}$C and $^{29}$Si NMR spectra could not be obtained. The electronic absorption spectrum is also not available for the same reason. Anal. Calcd for C$_{69}$H$_{69}$KN$_4$P$_2$Si$_2$Th: C, 61.68; H, 5.18; N, 4.17. Found: C, 59.94; H, 5.25; N, 3.85. $^1$H NMR (C$_6$D$_6$, 298 K): δ (ppm) 0.11 (s, 18 H, –SiMe$_3$), 3.93 (s, 1 H, –NHCPh$_3$), 6.89–7.07 (m, 22 H, ArH), 7.20–7.25 (m, 14 H, ArH), 7.56–7.58 (m, 6 H, ArH), 7.88–7.90 (m, 6 H, ArH). $^{31}$P NMR (C$_6$D$_6$, 298 K): δ (ppm) −2.24 (s). ATR-IR ν cm$^{-1}$: 3053 (w), 3020 (w), 2946 (w), 2893 (w), 1592 (w), 1483 (m), 1434 (s), 1352 (w), 1244 (s), 1103 (s), 1058 (s), 1025 (s), 893 (s), 828 (s), 749 (s), 697 (s), 633 (m), 594 (m), 546 (m), 523 (s), 510 (m), 480 (s), 414 (m).

**Preparation of [Th{C(PPh$_2$NSiMe$_3$)$_2$}(=NCPh$_3$)(κ$^2$-N, N′-2, 2-bipyridine)] (2ThBIPY).** At −78 °C, a solution of 2, 2′-BIPY (0.555 g, 3.55 mmol) and Ph$_3$CNH$_2$ (0.922 g, 3.55 mmol) in toluene (20 ml) was added into a stirring toluene solution of **3** (5.705 g, 5.93 mmol, 30 ml) in a dropwise manner. The addition was completed in 1 h. The red solution was allowed to gradually warm to ambient temperature and stirred at ambient temperature for 12 h then filtered. The deep red filtrate was concentrated to approximate 5 ml and stored at −35 °C overnight, affording **2ThBIPY** as a red crystalline solid (2.487 g, 54%). Anal. Calcd for C$_{60}$H$_{61}$N$_5$P$_2$Si$_2$Th: C, 59.94; H, 5.11; N, 5.82. Found: C, 60.29; H, 5.02; N, 5.45. $^1$H NMR (C$_6$D$_6$, 298 K): δ (ppm) −0.04 (s, 18 H, –SiMe$_3$), 6.87–6.91 (m, 2 H, ArH), 6.93–6.98 (m, 6 H, ArH), 7.02–7.15 (m, 9 H, ArH), 7.20–7.29 (m, 12 H, ArH), 7.40–7.44 (m, 4 H, Pyridine-H), 8.19–8.21 (m, 6 H, ArH), 8.23–8.28 (m, 4 H, Pyridine-H). $^{31}$P NMR (C$_6$D$_6$, 298 K): δ (ppm) −1.26 (s). $^{13}$C{$^1$H} NMR (C$_6$D$_6$, 298 K): δ (ppm) 3.29 (-SiMe$_3$), 21.77 (Th = N–CPh$_3$), 121.70, 124.64, 126.03, 127.46, 128.90, 129.45, 129.67, 130.61 (ArC), 131.91 (t, $^2J_{PC}$ = 6.0 Hz, C$_{meta}$ of P–Ph), 132.15 (t, $^2J_{PC}$ = 6.0 Hz, C$_{meta}$ of P–Ph), 138.22, 139.63, 152.20, 154.33, 156.45 (Pyridine-C). ATR-IR ν cm$^{-1}$: 3055 (w), 2941 (w), 2894 (w), 1591 (m), 1572 (w), 1478 (m), 1434 (s), 1309 (s), 1242 (s), 1113 (s), 1054 (s), 827 (s), 763 (s), 743 (m), 696 (s), 677 (s), 629 (s), 591 (s), 538 (s), 513 (s), 421 (s).

**Preparation of [Th(BIPM$^{TMS}$)(NHCPh$_3$)(CH$_2$SiMe$_3$)] (5).** At −78 °C a solution of Ph$_3$CNH$_2$ (136 mg, 0.9 mmol) in toluene (10 ml) was added slowly into a stirring solution of **3** (963.2 mg, 1 mmol) in toluene (10 ml). The mixture was stirred at −78 °C for 15 min, then at ambient temperature for 1 h. The mixture was then filtered, and all volatiles in the yellow filtrate were evaporated under vacuum to afford **5** as a yellow solid. Yield: 0.79 g, 95%. Complex **5** is a thermally unstable complex, decomposing in the solid state in a few days at −35 °C, or in a few hours in C$_6$D$_6$ solution at room temperature. Thus reliable microanalyses result, optical, IR, and $^{13}$C/$^{29}$Si NMR data are not available. $^1$H NMR (C$_6$D$_6$, 298 K): δ (ppm)

−0.06 (s, 2 H, -CH$_2$SiMe$_3$), 0.04 (s, 18 H, NSiMe$_3$), 0.49 (s, 9 H. –CH$_2$SiMe$_3$), 4.35 (s, 1 H, -NHCPh$_3$), 6.88–6.93 (m, 6 H, ArH), 7.20–7.26 (m, 9 H, ArH), 7.52–7.55 (m, 9 H, ArH), 7.70–7.81 (m, 6 H, ArH). $^{31}$P NMR (C$_6$D$_6$, 298 K): δ (ppm) 5.49 (s).

## Data availability
The X-ray crystallographic coordinates for structures reported in this Article have been deposited at the Cambridge Crystallographic Data Centre (CCDC), under deposition nos. 1861112-1861114. These data can be obtained free of charge from The Cambridge Crystallographic Data Centre (www.ccdc.cam.ac.uk/data_request/cif). All other data can be obtained from the authors on request.

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

## Acknowledgements

We thank the ERC (grant GoG612724), EPSRC (grants EP/P001386/1, EP/M027015, EP/N021932), Marie Curie Incoming Fellowship Scheme (grant 297888), and the University of Manchester for support, and the University of Manchester's Computational Shared Facility for computational resources. We are also grateful to the Pakistan HEC for IRSIP funding to S.S.

## Author Contributions

E.L. prepared the compounds and recorded and interpreted the characterisation data. S.S. and V.E.J.B. conducted and interpreted the theoretical calculations. N.K. directed and analysed the computational work and developed the central research idea. A.J.W. collected, solved, and refined the X-ray crystallographic data. S.T.L. originated and developed the central idea, analysed all the data, and wrote the manuscript with contributions from all co-authors.

## Additional information

**Competing interests:** The authors declare no competing interests.

