## [Peer Review File · Nature Communications]

Editorial Note: Parts of this peer review file have been redacted as we could not obtain permission to publish the reports of Reviewer 3.

Reviewers' comments:

Reviewer #1 (Remarks to the Author):

The manuscript "Emergence of the Structure-Directing Role of f-Orbital Overlap-Driven Covalency" discusses the role of the actinide (An) 5f and 6d orbitals for forming cis versus trans type of bonds associated also with transition from ionic to covalent bond. The preference between cis and trans geometry is investigated for carbene-carbene, carbene-imido and carben-oxo ligands for cerium, uranium and thorium. Using computational chemistry the authors show that the electrostatic potential surface around the metal center directs the cis geometry and not the 6d orbitals as it is currently believed. They also argue that the trans geometry becomes more favorable due to increasing participation of the 5f orbitals associated also with increasing overlap driven covalency of the metal-ligand bond. This 5f participation is a consequence of the short M-ligand bond length. The results demonstrate that the FEUDAL principle (the 5f orbitals are non-bonding) does not hold for many An systems and proposes a completely different way of understanding the trans bonding geometry, which is very typical for example for the actinyls. This investigation is performed by well-known world leaders in synthetic and computation actinide chemistry and surely is innovative. It helps to gain new insights into the complex electronic-structure-geometric constraints relationship of the actinide-ligand bonds. I recommend the manuscript for publication after revision.

Comments:

- Generally the 6d orbitals can also build covalent bonds. The authors however argue that the transition from cis to trans geometry is associated with change from 6d to 5f orbitals and from ionic to covalent bond. Could the authors discuss this point why they consider that the 6d orbitals form an ionic bond for these systems?
- Could the authors explain more in details the definition of the electrostatic potential surface around the An atom and the natural molecular orbital approach. Also, it will be useful for the general reader if they cite other relevant An studies where analyses of those were applied.
- It will be helpful if the different atoms are indicated in Figure 5.
- The authors write:

"The NLMO shown in Figure 5 is highly directional, and would clearly disfavour an incoming negatively charged ligand in the trans position, i.e. it likely accounts for the cis-directing ESP."

This might be clear from Fig. 5 for a computational chemist but not really for the general reader. The authors might think how to improve the presentation of this figure and or the complement the text. I think that it will help if you add Th=C and BIPM over the ESP blue surfaces (name them). Then it will be more clear that Fig. 5 shows only the spherical ESP surface of Th=C from Fig. 4. This is at least how I understand it.

- Could you give the correlation from Fig. 6 individually for each metal in the Si? This will be a very useful information.
- In the text:
"..very little correlation of the [C]=N=E angle with metal d-character to the NLMO σ -bonding.."
[C]=N=E maybe should be [C]=M=E ?
- Could the authors define TSs in the manuscript?
- Could the authors comment on the definition of the R2 parameter?
- It is generally accepted that the covalency of the actinide-ligand bond increases when the atomic overlap of orbitals increases or the energy difference between the metal and ligand valence orbitals decreases. The authors point out that they mean the former "type" of covalency change and estimate it by looking at the 5f/6d contribution to specific orbitals. Is this contribution a measure of the orbital-overlap driven covalency? It will be useful to clearly

point out this since there is a discussion going on in the literature on this topic. Can you please comment how the energy match driven covalency changes for Th compared to U? Is it possible to derive this from the calculations you have performed?

- Could the authors comment on the role of the semi-core U 6p orbital for the cis versus trans geometries? How its contribution changes as a function of the bonding angle (Fig.6)? Such analysis will be surely useful also for the 6d orbitals.

- Could you specify for which geometries is exactly the data in Table 1?

- Could the authors analyze the changes of the ESP for C-U-O (trans geometry) similarly they do for the Th cis? They do say that it behaves differently.

- I am surprised that the authors do not cite:

To cite this article: Kenneth G. Dyall (1999) Bonding and bending in the actinyls, *Molecular Physics*, 96:4, 511-518, DOI: [10.1080/00268979909482988](https://doi.org/10.1080/00268979909482988)

- It will be also useful to cite a recent article discussing a new way to spectroscopically detect elongation and bending of the actinyl bonds

T. Vitova et al. *Inorg. Chem.* 2018, 57, 4, 1735-1743.

Reviewer #2 (Remarks to the Author):

Inverse-trans-influence (ITI) is a case in metal-ligand binding using 6d-orbitals without 5f-orbital participation in actinide chemistry. In this work, the authors synthesized a series of special M- FEUDAL complexes (M= Ce-, Th, and U, ligands=NHRK, and BIPY). NLM, QTAIM, and some other theoretical methods reveal that these M-C, M-N bonds were dominated by covalent interaction, and was not only deduced from d-orbital, but also orbital overlap driven f-orbital of actinides, which greatly varied with the previous conclusion of these FEUDAL (f's essentially unaffected, d's accommodate ligands). Therefore, this is an very exciting finding in actinide coordination chemistry. This work is well conducted, and presented, and has much importance for deep understanding of covalent bonding characters of actinides complexes. I recommend accept after some minor corrections.

(1) The authors had better give a schematic of the cis-and trans-geometry of one complex they studied in the manuscript or ESI to make it better understood. This helps to understand the statement "NLMO has significant amplitude in the trans direction, resulting in a cis directing effect" as discussed in the section "NLMO" and some descriptions in the section "Rationalising a Model for the Interplay of Electrostatics and Covalency"

(2) In previous work (*Dalton Trans.* 2018, 47, 12718), the four functionals were used to evaluate the reliability of structural optimization for the uranium-biscarbene complex, it was found that GGA BP86 functional are the best choice to match available experimental values. So the authors the authors had better cite this work.

(3) It is difficult to understand that "the ESP surface around the metal is asymmetric" and the authors had better give some explanations.

(4) I am very curious why the author can draw the following conclusions in page 8. "Comparison of these at the optimised geometries with those of the structures at the end of the energy scans reveals that the σ -orbitals exhibit the greatest energy changes; at least twice those found for the π -orbitals and more often substantially greater ($> \times 20$). Additionally, the σ -orbitals exhibit the greater change in contribution from the metal; on average, these changes are an order of magnitude larger in the σ - than the π -orbitals." Please provide more explanation or cite related works to support this point.

(5) Why the authors choose the element of Ce instead of other elements as a reference?

(6) In page 9, "Extending this argument, elongating the M-O distance in either [C]=Ce=O or [C]=U=O should reduce the M-O interaction and favour a smaller C-M-O angle." of "favour a smaller C-M-O" is not easy to understand, and need a reorganization.

(7) In page 22, in the step of synthesizing compound 5 and 2THBIPY from 3, the material Ph₃NH₂ is added to 0.9 eq to obtain the products, why are they not reacted in a stoichiometric ratio? I found that the ratio given in the support information is inconsistent with that in Fig 1.

(8) In support information, the details for the synthesis of compound 5 are not included. More details are expected.

(9) In page 25, Fig 5 should be reorganized, and add the atomic legend to make it more readable.

(10) In support information, Fig S1 and S8, the other undefined peak should be attributed. And in Fig S4, S7 and S11, the characteristic peak of the functional group should be simply identified. There is a few of minor comments

1. In page 7 Line5: "(Figure 3b)" to "(Figure 3c)".
2. In page 8 Line6: "[[C]=U]2+" to "[[C]=Th]2+".
3. In page 9 Line5: "[C]=N=E" to "[C]=M=E".
4. In page 6, "SCF" must be given clearly definition.

[Redacted]

Reviewer Comments and Responses:

Reviewer #1:

The manuscript "Emergence of the Structure-Directing Role of f-Orbital Overlap -Driven Covalency" discusses the role of the actinide (An) 5f and 6d orbitals for forming cis versus trans type of bonds associated also with transition from ionic to covalent bond. The preference between cis and trans geometry is investigated for carbene-carbene, carbene-imido and carben-oxo ligands for cerium, uranium and thorium. Using computational chemistry the authors show that the electrostatic potential surface around the metal center directs the cis geometry and not the 6d orbitals as it is currently believed. They also argue that the trans geometry becomes more favorable due to increasing participation of the 5f orbitals associated also with increasing overlap driven covalency of the metal-ligand bond. This 5f participation is a consequence of the short M-ligand bond length. The results demonstrate that the FEUDAL principle (the 5f orbitals are non-bonding) does not hold for many An systems and proposes a completely different way of understanding the trans bonding geometry, which is very typical for example for the actinyls. This investigation is performed by well-known world leaders in synthetic and computational actinide chemistry and surely is innovative. It helps to gain new insights into the complex electronic-structure-geometric relationship of the actinide-ligand bonds. I recommend the manuscript for publication after revision.

Comments:

Generally the 6d orbitals can also build covalent bonds. The authors however argue that the transition from cis to trans geometry is associated with change from 6d to 5f orbitals and from ionic to covalent bond. Could the authors discuss this point why they consider that the 6d orbitals form an ionic bond for these systems?

RESPONSE: The reviewer is correct that we do argue a change from ionic to covalent bonding but we are not arguing a transition from 6d to 5f linked in such a direct trade-off. We therefore feel this point is not appropriate to comment on.

Could the authors explain more in details the definition of the electrostatic potential surface around the An atom and the natural molecular orbital approach. Also, it will be useful for the general reader if they cite other relevant An studies where analyses of those were applied.

RESPONSE: Yes, we have now provided additional information in this section at the top of page 8. It will be helpful if the different atoms are indicated in Figure 5.

RESPONSE: We have added labels to the figure.

The authors write: "The NLMO shown in Figure 5 is highly directional, and would clearly disfavor an incoming negatively charged ligand in the trans position, i.e. it likely accounts for the cis-directing ESP." This might be clear from Fig. 5 for a computational chemist but not really for the general reader. The authors might think how to improve the presentation of this figure and or the complement the text. I think that it will help if you add Th=C and BIPM over the ESP blue surfaces (name them). Then it will be more clear that Fig. 5 shows only the spherical ESP surface of Th=C from Fig. 4. This is at least how I understand it.

RESPONSE: We have added additional text wrt figure 5 on page 8. Figure 5 does not show only the spherical ESP surface. Rather, it provides an orbital-based explanation as to why the ESP in Figure 4e is asymmetrically oriented in the cis position.

Could you give the correlation from Fig. 6 individually for each metal in the Si? This will be a very useful information. RESPONSE:

We have added this to the figure 6 caption.

In the text: "...very little correlation of the [C]=N=E angle with metal d-character to the NLMO σ -bonding.." [C]=N=E maybe should be [C]=M=E ?

RESPONSE: Good spot, yes correct and corrected. Could

the authors define TSs in the manuscript?

RESPONSE: Yes, this is a good suggestion and we have now done this at the top of page 6.

Could the authors comment on the definition of the R2 parameter?

RESPONSE: We don't feel that we should have to define something standard from a least squares regression analysis in a research paper.

It is generally accepted that the covalency of the actinide-ligand bond increases when the atomic overlap of orbitals increases or the energy difference between the metal and ligand valence orbitals decreases. The authors point out that they mean the former "type" of covalency change and estimate it by looking at the 5f/6d contribution to specific orbitals. Is this contribution a measure of the orbital-overlap driven covalency? It will be useful to clearly point out this since there is a discussion going on in the literature on this topic. Can you please comment how the energy match driven covalency changes for Th compared to U? Is it possible to derive this from the calculations you have performed?

RESPONSE: We have added text to page 9 clarifying the issue of orbital -overlap. We believe this does reflect what we assert because energy- driven covalency emerges as an important effect in the actinide series only for elements heavier than Th and U, most notably for the minor actinides Am and Cm. It will be at most a minor component in Th and U bonding, and changes little between them. It would not be straightforward to determine from the present calculations, nor is it necessary.

Could the authors comment on the role of the semi-core U 6p orbital for the cis versus trans geometries? How its contribution changes as a function of the bonding angle (Fig.6)? Such analysis will be surely useful also for the 6d orbitals.

RESPONSE: The 6p and 6d orbital contributions to the σ -bonding NLMOs at the optimized geometries are given in Table 1, as are the 5f, on which Figure 6 is based. We explicitly state in the paper that there is very little correlation of 6d character with [C]=M=E angle ($R^2=0.36$). Table 3 reveals that the 6p character is very small, and hence it cannot play a role in determining cis vs. trans geometries.

Could you specify for which geometries is exactly the data in Table 1?

RESPONSE: Yes, we have added clarifying text to the caption of Table 1.

Could the authors analyze the changes of the ESP for C-U-O (trans geometry) similarly they do for the Th cis? They do say that it behaves differently.

RESPONSE: We do not understand the question and wonder if the reviewer has misunderstood. The ESP is analysed in the absence of an E ligand (e.g. the O atom referred to by the reviewer here), only looking at the approach of the BIPM carbene ligand towards the metal. Hence we do not analyse the ESP at a cis or trans geometry as they are not defined in the absence of an E ligand. We have previously analysed the ESP for [C]=U²⁺ - see reference 63 as we state in the paper. Its change as a function of C-U distance is very similar to that shown for [C]=Th²⁺ in Figure 4.

I am surprised that the authors do not cite: Kenneth G. Dyall, 1999, Bonding and bending in the actinyls, *Molecular Physics*, 96:4, 511-518. It will be also useful to cite a recent article discussing a new way to spectroscopically detect elongation and bending of the actinyl bonds T. Vitova et al. *Inorg. Chem.* 2018, 57, 4, 1735-1743.

RESPONSE: we have added both references as 49 and 50 in the revised manuscript.

Reviewer #2:

Inverse-trans -influence (ITI) is a case in metal-ligand binding using 6d-orbitals without 5f-orbital participation in actinide chemistry. In this work, the authors synthesized a series of special M-FEUDAL complexes (M= Ce-, Th, and U, ligands=NHRK, and BIPY). NLM, QTAIM, and some other theoretical methods reveal that these M-C, M-N bonds were dominated by covalent interaction, and was not only deduced from d-orbital, but also orbital overlap driven f- orbital of actinides, which greatly varied with the previous conclusion of these FEUDAL (f's essentially unaffected, d's accommodate ligands). Therefore, this is an very exciting finding in actinide coordination chemistry. This work is well conducted, and presented, and has much importance for deep understanding of covalent bonding characters of actinides complexes. I recommend accept after some minor corrections.

The authors had better give a schematic of the cis-and trans-geometry of one complex they studied in the manuscript or ESI to make it better understood. This helps to understand the statement "NLMO has significant amplitude in the

trans direction, resulting in a cis directing effect” as discussed in the section “NLMO” and some descriptions in the section “Rationalising a Model for the Interplay of Electrostatics and Covalency”

RESPONSE: We have amended chart 1 to include pictures of the cis and trans geometries of the models we investigate computationally

In previous work (Dalton Trans. 2018, 47, 12718), the four functionals were used to evaluate the reliability of structural optimization for the uranium-biscarbene complex, it was found that GGA BP86 functionals are the best choice to match available experimental values. So the authors had better cite this work.

RESPONSE: This paper is now cited as reference 72.

It is difficult to understand that “the ESP surface around the metal is asymmetric” and the authors had better give some explanations.

RESPONSE: In response to reviewer 1 we have added some qualifying text at the bottom of page 8 with respect to the ESP, and it is quite clear from Figure 4 that the ESP around the Th is essentially spherical in image a but much less so in image e. We feel these two things together now make it clear.

I am very curious why the author can draw the following conclusions in page 8. “Comparison of these at the optimised geometries with those of the structures at the end of the energy scans reveals that the σ -orbitals exhibit the greatest energy changes; at least twice those found for the π -orbitals and more often substantially greater ($> \times 20$). Additionally, the σ -orbitals exhibit the greater change in contribution from the metal; on average, these changes are an order of magnitude larger in the σ - than the π -orbitals.” Please provide more explanation or cite related works to support this point.

RESPONSE: The σ component is much the dominant component of the energy change. It is possible that the π plays a minor role, but not enough to change the conclusions. The data has been added in Table S3 of the SI and reference is added to the text on page 9.

Why the authors choose the element of Ce instead of other elements as a reference?

RESPONSE: It is very common in early actinide studies, both experimental and computational, to make comparisons with the early lanthanide elements. As Ce is the only early lanthanide with a +4 oxidation state, and indeed Ce(IV) complexes with BIPM have been made (references 60-62 in the revised manuscript), it is the only logical element to choose for such a comparison.

In page 9, “Extending this argument, elongating the M-O distance in either [C]=Ce=O or [C]=U=O should reduce the M-O interaction and favour a smaller C-M-O angle.” of “favour a smaller C-M-O” is not easy to understand, and need a reorganization.

RESPONSE: We feel the reviewer must have misread this as it makes sense and our English is correct and we will not change it.

In page 22, in the step of synthesizing compound 5 and 2THBIPY from 3, the material Ph₃NH₂ is added to 0.9 eq to obtain the products, why are they not reacted in a stoichiometric ratio? I found that the ratio given in the support information is inconsistent with that in Fig 1.

RESPONSE: As we pointed out in the figure 1 caption the sub-stoichiometric amount is to suppress ligand scrambling reactions. However, the reviewer is correct there was a mistake in the ratios, which has now been corrected for 2ThBIPY.

In support information, the details for the synthesis of compound 5 are not included. More details are expected.

RESPONSE: Complex 5 is thermally unstable but we have been able to acquire limited spectroscopic data, which we include along with the experimental preparation and with the addition of two figures in the SI of ¹H and ³¹P NMR spectra.

In page 25, Fig 5 should be reorganized, and add the atomic legend to make it more readable. RESPONSE:

We have added labels to the figure to make it clearer.

In support information, Fig S1 and S8, the other undefined peak should be attributed. And in Fig S4, S7 and S11, the characteristic peak of the functional group should be simply identified.

RESPONSE: Good spot. We have updated the relevant NMR figures with attributions now. For IR, (Fig S4, S7 and S11), the absorptions at $\sim 3000/\text{cm}$ are C-H stretching and not of chemical significance, while the major functional groups, i.e. M=C and P=N, are overlapped by other peaks in the fingerprint region and unidentifiable.

There is a few of minor comments

1. In page 7 Line5: "(Figure 3b)" to "(Figure

3c)". RESPONSE: correction made.

2. In page 8 Line6: "[[C]=U]2+" to "[

[[C]=Th]2+". RESPONSE: correction made.

3. In page 9 Line5: "[C]=N=E" to "[C]=M=E".

RESPONSE: correction made.

4. In page 6, "SCF" must be given clearly

definition. RESPONSE: correction made.

[Redacted]

REVIEWERS' COMMENTS:

Reviewer #1 (Remarks to the Author):

I agree with the answers and the corrections and have no more comments.

Reviewer #2 (Remarks to the Author):

The authors have fully addressed my concerns, now I'd like to suggest accept.

[Redacted]